# Rationalisation of the UK Nutrient Databank for Incorporation in a Web-Based Dietary Recall for Implementation in the UK National Diet and Nutrition Survey Rolling Programme

**DOI:** 10.3390/nu14214551

**Published:** 2022-10-28

**Authors:** Birdem Amoutzopoulos, Toni Steer, Caireen Roberts, David Collins, Kirsty Trigg, Rachel Barratt, Suzanna Abraham, Darren James Cole, Angela Mulligan, Jackie Foreman, Anila Farooq, Polly Page

**Affiliations:** Nutrition Measurement Platform, MRC Epidemiology Unit, University of Cambridge, Cambridge CB2 0QQ, UK

**Keywords:** 24 h recall, dietary assessment, rationalisation, food composition, NDNS, Intake24, UK Nutrient Databank, evaluation, nutrient intake, food database

## Abstract

The UK National Diet and Nutrition Survey rolling programme (NDNS RP) commenced in 2008 and moved in 2019 from a traditional paper food diary to a web-based 24 h recall, Intake24. This paper describes the approach to update and downsize the underlying UK Nutrient Databank (NDB) for efficient data management and integration into Intake24. Consumption data from the first 10 years (2008/2009 to 2017/2018) of NDNS RP informed decisions on whether foods from the extensive UK NDB were to be retained, excluded, revised or added to for creation of a rationalised NDB. Overall, 5933 food codes in the extensive NDB were reduced to 2481 food codes in the rationalised NDB. Impact on assessment of nutrient intakes was evaluated by re-coding NDNS 2017 data using the rationalised NDB. Small differences were observed between estimated intakes (Cohen’s *d* ≤ 0.1) for all nutrients and there was a good level of agreement (Cohen’s κ ≥ 0.6) between the extensive and rationalised NDBs. The evaluation provides confidence in dietary intake estimates for ongoing nutritional surveillance in the UK and strengthens the evidence of a good agreement between concise food databases and large food databases incorporated into web-based 24 h recalls for estimating nutrient intakes at the population level.

## 1. Introduction

Web-based, self-administered 24 h dietary recalls represent a useful digital tool enabling efficient and cost-effective dietary assessment, providing relatively detailed quantitative dietary intake data with minimal bias [1,2,3,4,5,6,7,8]. To minimise administration burden on respondents and enable efficient self-completion, such 24 h recall systems require manageable food and nutrient databases and a simple user interface [3,9].

The UK National Diet and Nutrition Survey Rolling Programme (NDNS RP, 2008-) [10] is a continuous running cross-sectional survey assessing food consumption and nutrient intakes in a nationally representative sample of more than 1000 respondents aged 1.5 years and over, living in private households in the UK each year. The survey is funded by UK government, initially Public Health England (PHE) and the UK Food Standards Agency, and from October 2021 responsibility and funding transferred from PHE to the Office for Health Improvement and Disparities at the Department for Health and Social Care. For the first eleven years of the rolling programme, a paper food diary was used to collect dietary data. Foods were reported by participants using free text format and paper diaries were subsequently manually coded by trained research assistants, linking to a specially maintained food composition database: UK Nutrient Databank (NDB). Complex foods were largely coded at ingredient/recipe level. This process was time-consuming and expensive, impacting timeliness of data availability and curtailing potential to scale the survey. With advances in technology-based methods offering potential for more efficient, cost-effective and less burdensome dietary assessment, a review of available computer/digital tools was undertaken in 2017/2018 to identify a new method for NDNS RP [11]. Intake24, the online 24 h recall tool originally developed by Newcastle University and Food Standards Scotland [12], was ultimately selected by the NDNS Project Board to replace the paper food diary.

In order to implement Intake24 in the NDNS RP, the tool required some technical adaptation and customisation to meet specific survey requirements. The embedded food databases (i.e., food list, portion information, associated food probes) and NDB also needed updating. An instance of Intake24, based on Intake24 Newcastle 2014 (Software Version 3, incorporating UK NDB-2012, NDNS Year 4) was built by the University of Cambridge, working in collaboration with the University of Newcastle for technical leadership. Following tool development and implementation, data collection using Intake24 commenced in NDNS RP in October 2019 using https://intake24.org/ (Software Version 3: Intake24 [UK] Cambridge 2019, accessed on 27 June 2022).

The UK NDB contains a wide range of food codes, including single foods, manufactured products, homemade recipe dishes, and dietary supplements. The nutrient composition values in the NDB are based on nutrient analysis of foods [13] and have been updated using manufacturers’ data gathered through food labels as well as data from the scientific literature, non-UK food composition tables and recipe calculations. To date for the NDNS, NDB updates have been carried out annually with priorities based on reported consumed foods for each survey year, and to reflect changes in the UK food system. As a result, at the time of implementing Intake24 in the NDNS, the UK NDB (“extensive NDB”) was of considerable size (*n* = 5933) and had grown unsystematically with duplicative and redundant foods. Hence, it was not fit-for-purpose for integration into Intake24. Furthermore, there were differences in the way recipes were reported and coded between the paper diary and Intake24 (in the latter the coding structure was moved from ingredient coding to generic reference food codes) and the NDB needed to be adapted to accommodate this difference. Therefore, the NDB required both a significant update and downsizing to enable efficient data management and application within Intake24. This paper describes the method devised to comprehensively and systematically review and rationalise the extensive NDB with the objective to reduce its size and generate a more concise NDB (“rationalised NDB”). Given the role of NDNS data as an Official Statistic for UK national nutritional surveillance, it was also important to evaluate the impact of changing the dietary assessment methodology in the survey; this included changes in the underpinning NDB in order to assess any impact on dietary data and continuity of trend analyses. This paper also presents results on the impact of using the rationalised NDB on NDNS RP data compared to the use of the former, more extensive NDB.

## 2. Materials and Methods

### 2.1. Rationalisation of the UK NDB

Each food in the extensive NDB (*n* = 5933) was independently reviewed by two research assistants to identify foods relevant for inclusion in the rationalised NDB. The review included consideration of key factors (Figure 1) including reported consumption frequency in first 10 years of NDNS RP (2008/2009 to 2017/2018), public health policy and monitoring focus, and details and coverage of foods/products. Consumption data from the first 10 years of NDNS RP informed decisions on whether foods from the extensive UK NDB were to be retained, excluded, revised or added to for creation of a rationalised NDB. All available data from the first 10 years were used where feasible, but where not feasible/possible (e.g., for reasons of resource, time, complexity, practicality), data were taken from most recent year available (2017).

To enable comprehensive review, foods were examined by NDNS food group [15], thereby taking into account related foods. For example, decisions on fruit yogurt were made by considering in relation to the yogurt group as a whole. 

Using a broad “food matching” process, all foods and dietary supplements in the extensive NDB were reviewed to identify whether they should be retained, excluded or could be adequately represented using another food code. See Appendix A for an illustration of the food matching method. A cut-off approach was used to compare the nutrient composition for consideration of similar foods. Using the integrated dietary assessment system, Diet In Nutrients Out (DINO) [16], the composition of foods was compared in pairs using cut-offs, key nutrients, and the average amount of foods consumed by 1 to 3 year olds in NDNS year 2017 (representing the youngest age group in NDNS to retain maximum sensitivity in the food reference database and using the most recent survey year available at the time). The steps followed in this matching process are explained through a yogurt example in Appendix A. 

For each food in the extensive NDB, one of the following categories were assigned:

RETAIN: Foods to be retained in the rationalised NDB: Frequently consumed foods and/or foods that could not be adequately represented (either according to their food name or nutrient profile) by other foods existing in the extensive NDB were marked to retain. Some foods were retained where consumption was infrequent due to other considerations. For example, fortified milk was retained although it was consumed only 3 times between 2008 to 2018, owing to its fortificant profile. Other infrequently consumed foods were also retained if they were a specific traditional/ethnic food (e.g., Turkish delight, halva) or a specific branded food with food policy interest.

EXCLUDE: Foods in the extensive NDB to be excluded from the rationalised NDB: These were mostly infrequently consumed foods (i.e., consumed less than 10 times in NDNS years 2008–2018). Among the excluded foods (*n* = 3602), the average frequency of consumption was total 6 times in 10 years. For example, béarnaise sauce was not reported between 2008 to 2018, therefore it was excluded.

REPRESENT: Foods in the extensive NDB that could be represented by another food: Similar foods with a similar nutrient composition profile were represented by another code. Similarity was assessed either objectively through nutrient comparisons (“food matching”) and/or according to principles based on public health priorities. For example, the distinction of ultra-high temperature (UHT) vs. pasteurised milk was considered unnecessary for the rationalised NDB so UHT milk was represented by pasteurised milk; the differentiation of canned vs. non-canned fish was no longer necessary for exposure assessment, therefore “canned crab” was represented by “boiled crab”. 

NEW/REVISED: New generic food codes to be created, food composition codes to be revised and/or updated for the rationalised NDB. 

The extensive NDB codes were extensively revised/updated or new codes were added to allow for a single code to represent a range of similar foods thereby enabling reduction of number of food codes overall. This approach was also used to improve efficiency and to enable implementation of a single food code for common dishes where ingredients may have been previously coded separately as a recipe. A sub-sample of mixed dish codes was selected for revision based on their priority according to the following criteria: (a) dishes reported more than 20 times in NDNS year 2017 (only one year was used as recipes were more easily identifiable in NDNS year 2017 dataset following a change in coding procedures) and (b) the difference in energy between actual reported dishes and matched codes in the extensive NDB being larger than +/−20kcal in 100 g edible portion. Single codes for common dishes were identified by the following steps: Individual recipes reported in NDNS RP year 2017 were assigned to a generic dish based on name of recipe (data were compiled in a Microsoft (MS) Access Database). For example, all varieties of fried bacon such as “bacon fried in sunflower oil” and “bacon fried in olive oil” were grouped into “bacon fried”. This exercise used recipe data from only one year of NDNS RP as the ingredients of recipes were more easily identifiable in the NDNS year 2017 dataset due to coding approaches.Common ingredients and median energy content of each dish group (e.g. varieties of fried bacon) were matched to a food code available in the extensive NDB or to a new food code created for the rationalised NDB (see Appendix A for illustration example on the list of fried bacon reported in NDNS year 2017 which were ultimately matched to a single fried bacon food code).The rationalisation exercise was designed to produce a rationalised NDB for the UK, comprising a comprehensive list of current foods each corresponding to an associated nutrient composition code. In parallel with 2019–2020 NDNS data collection using https://intake24.org/ (Software Version 3: Intake24 [UK] Cambridge 2019, accessed on 27 June 2022), data checks were performed on collected dietary intake data in order to review and monitor suitability and coverage of food codes and to identify further refinements.

### 2.2. Evaluation of Rationalisation

The impact of using the rationalised NDB for collection of dietary data in the NDNS RP was evaluated through a modelling and recalculation exercise on NDNS year 2017 data, to compare the use of the rationalised NDB in place of the extensive NDB. Food codes from NDNS year 2017, originally collected using the paper food diary and manually coded using the extensive NDB, were matched to food codes available in the rationalised NDB using the following method.

Step 1 Single food match: Single foods reported as consumed in NDNS year 2017 were matched to suitable food codes available in the rationalised NDB. 

Step 2 Recipe match: The combinations of foods (e.g., mostly ingredients of homemade dishes or ready meals) that were previously coded as multiple ingredients to make up a dish in NDNS year 2017 were matched to suitable corresponding single food codes now available in the rationalised NDB for respective dishes. To manage the considerable variation in ingredient profiles resulting from free text reporting and manual coding methods of coded recipes in the NDNS year 2017 dietary dataset, recipes were first grouped according to their ingredients and then manually matched to a food code existing in the rationalised NDB. For example, omelette ingredients (e.g., egg, oil, cheese) were matched to the cheese omelette code in the rationalised NDB. Due to complexity of matching and a large number of recipes (*n* = approx. 5000) reported in NDNS year 2017, a pragmatic matching approach based on similarity in food names was adopted. The food groups of ingredients were combined in an Excel file to group similar recipes to make better matches (Appendix A provides an example on recipe matching).

A quality review was carried out including checks for recipe replacements where differences in energy, meat, and fruit and vegetable proportions between the extensive and the rationalised NDB codes were the greatest and consumption rates were relatively high. Following this review, some of the food codes in the rationalised NDB were subsequently refined (*n* = 33) and/or recipes in NDNS year 2017 were relinked to more suitable food codes. For example, the composition of chicken curry in the rationalised NDB was amended by adding more vegetables.

Step 3 Sandwich match: Foods that were identified as consumed as part of a sandwich in NDNS year 2017 were matched to a suitable single sandwich food code available in the rationalised NDB following a standard protocol (Appendix A). Due to former coding protocols, foods consumed as sandwiches were not always readily identifiable as sandwiches in the NDNS year 2017 dataset. For example, bread, butter, cheese may not have been linked together as components of cheese sandwich in the NDNS year 2017 dataset, therefore where the bread, butter and cheese were reported together in the same time slot, they were judged to constitute a cheese sandwich and were named and matched accordingly. Due to a large number of assumed sandwiches (*n* = 4915), most sandwiches were matched based on assigned food name as opposed to matching them based on detailed nutrient/recipe profile. 

After the matching process, the 2017 dietary data were recalculated using the rationalised NDB. 

### 2.3. Statistical Analysis for the Evaluation 

The statistical analysis in this study was informed by the approach used by Evans’ et al [17] in similar work to develop and evaluate a concise food list for Foodbook24, another web-based 24 h dietary tool. In our study, estimates of intake for selected foods and nutrients (energy, proximates, vitamins, minerals, meat, fish, fruit and vegetable) were recalculated following matching using the rationalised NDB, and compared to the original NDNS year 2017 intake data using paired-samples t tests, performed on the raw or log transformed scale as appropriate, to provide *p*-values. For consistency, all differences are provided as percentage differences. Cohen’s *d* [18] effect size was used to examine differences between intake estimates using the extensive NDB and the rationalised NDB to provide an indication of significance between the two estimates. A Cohen’s *d* effect size below 0.2 or above −0.2 can be interpreted as having little significance and no meaningful effect (given the small effect size in relation to the observed variability).

To further interpret differences, a cross-classification analysis was undertaken to quantify the level of agreement between categorisation of estimates into tertiles of intake distribution. The percentage of NDNS year 2017 respondents categorised into the same tertile of intake in the two NDB scenarios was calculated and the level of agreement quantified using Cohen’s κ statistic. A Cohen’s κ statistic above 0.8 implies very good agreement and above 0.6 implies good agreement between the two variable measures. 

Cross-classification analysis was also used to quantify the level of agreement (described by Cohen’s κ) in the percentages of respondents meeting key nutritional recommendations on a subset of the selected foods and nutrients (total fat, saturated fat, carbohydrate, dietary fibre, free sugars and 5 A Day portions of fruit and vegetables) for both database scenarios. 

All statistical analyses were performed using R version 3.6.1 (https://www.R-project.org/, accessed on 27 June 2022).

## 3. Results

### 3.1. Rationalisation and Update of NDB

Using the criteria described in the methods section (under: 1. Rationalisation of the UK NDB), 2331 codes in the extensive NDB were assigned to RETAIN (39%), 1541 codes to EXCLUDE (26%) and 2061 codes to REPRESENT (35%). For NEW/REVISED, 150 new generic food codes were created and the food composition of 35 existing food codes (mostly mixed dishes or their components) were amended. New codes added were mostly for sandwiches and salads that were previously coded as individual components. Overall the resulting rationalised NDB contained 2481 foods (including the addition of new foods), representing an overall reduction of 58% compared to the extensive NDB (*n* = 5933).

### 3.2. Evaluation of Change in Dietary Data Output

For the evaluation exercise, overall, 49% of foods consumed in NDNS year 2017 were re-matched to a different code in the rationalised NDB, of which most were replaced with a generic recipe (mixed dish) code (25%) or single food codes (20%) followed by sandwich codes (4%) (Table 1). The remaining 51% foods were unchanged because the original food code had not been affected by the rationalisation process. There were greater nutrient differences (percentage difference between mean nutrient intake) in mixed dishes (e.g., lasagne) (differences ranged from −9% to 27%) compared with single foods (e.g., milk) (differences ranged from −4% to 0.7%) and sandwiches (differences ranged from −5% to 0.6%).

Mean daily intake coded using the two NDB versions (extensive and rationalised NDB) was compared for twenty-four nutrients (Table 2) and three foods (fruit and vegetable, meat, and fish) using the NDNS year 2017 data (all ages combined). The significance of the difference between estimated intakes was small (i.e., ≤0.2 defined by Cohen’s *d*) for twenty-four nutrients and for three foods (fruit and vegetable, meat, and fish) (Table 2). The percentage difference was smaller for energy (1.1%) and macronutrients (ranging from 0.0% to 3.1%) compared to micronutrients (ranging from 0.3% to 9.6%). The difference was greater than 5% for vitamins D and E. Larger percentage differences (between −4.4% and 5.7%) were observed for each of the three foods (fruit and vegetable, meat, and fish) although still relatively small when significance was measured based on Cohen’s *d* ≤ 0.2. When nutritional supplements were included, the differences in mean estimated nutrient intakes between the two NDBs were also relatively small (Cohen’s *d* ≤ 0.2) (Appendix A). 

There was a good level of agreement according to the size of Cohen’s κ (i.e., ≥0.6) for all nutrients and foods in relation to both the comparison of nutrient and food intake estimates and percentage of respondents meeting nutrition recommendations using the two NDB scenarios (Table 3 and Table 4). Over 80% of respondents were classified into the same tertile of intake for the majority of nutrients (21 out of 24 nutrients) using the rationalised NDB compared to the extensive NDB. The exception was for vitamins E (73.6%), A (78.8%), and D (73.6%) and %TE (total energy) protein (78.7%), %TE carbohydrate (78.3%) and %TE saturated fat (78.4%) (Table 3). Over 90% of respondents were classified into the same tertile for energy, carbohydrate, fruit and vegetable, and fish (Table 3). When nutritional supplements were included, there was a good level of agreement for selected nutrients according to the size of Cohen’s κ (≥0.6) (Appendix A).

For carbohydrate, free sugars, total fat, saturated fat, fibre, and fruit and vegetable intakes, over 80% of respondents were classified into the same categories of nutrient adequacy using the rationalised NDB compared to the extensive NDB (Table 4). The highest proportion (17%) of respondents who moved categories of nutrient adequacy were for total fat (% TE). This proportion is equally split with 9% moving into the recommendation range (20–35% TE) after rationalisation and 8% moving out of this range after rationalisation. There was no clear association with age or sex for moving/not moving categories. There was a slightly higher than expected (according to NDNS year 2017 overall) proportion of males who stayed in the 20–35% TE range (50% of respondents who stayed in the 20–35% TE range were male vs. 45% were male overall) and a slightly older than expected (according to NDNS year 2017 overall) age for those who moved into the 20–35% TE range (the average age was 36yrs for those who moved into the 20–35% TE range vs. average age of 30 years overall) but these differences were very minor. Appendix A shows that the respondents changing categories were very close to the boundary even though they were not in the 20–35% TE range which overall suggests no loss of precision following rationalisation.

## 4. Discussion

Our paper adds to the body of food composition literature, providing methodology to identify and remove redundant coding detail in nutrient composition databases to achieve a more proportionate generic coding frame. Our adaptation and integration of the UK NDB into Intake24 has produced a standardised and systematic rationalisation of the former extensive NDB, enabling more efficient database management and use of a generic food list with minimal impact on resulting dietary intake data. Other research groups [8,17,23,24] developing/using existing technology-based 24 h recalls in a new setting similarly cleaned or updated the underlying food databases as a first step of method development or update. Evans et al. [17] devised a concise food list (*n* = 2319) to be used in a self-administered web-based 24 h dietary recall, Foodbook24, for use in Ireland. Koch et al. [8] also updated the German Food Code and Nutrient Data Base (Bundeslebensmittelschlüssel (BLS) version 3.02) as part of the adaptation process of myfood24 Germany, a web based 24 h dietary recall, for use in German populations. They followed an editing process, which included steps such as removing or merging foods, reducing the number of foods from 15,000 to 7177, where the proportion of reduction in the original food database (52%) was similar to the reduction in our rationalisation of the extensive NDB (58%). On the other hand they subsequently combined this database with a large brand database (*n* = 4324). Adler et al. [25] followed a systematic approach to review and rationalise the foods that existed in the USDA Food and Nutrient Database (*n* = approx., 8536 [26]) used in What We Eat in America (WWEIA) and National Health and Nutrition Examination Survey (NHANES). Ultimately, they discontinued 279 foods and added 1200 foods whereas our implementation discontinued 3602 foods and added 150 foods.

This paper provides detail on all steps of our rationalisation process and includes methods not previously described by others. We used recent representative UK consumption data from the NDNS RP as the reference base to inform understanding of common foods and consumption frequency and devised a framework for consideration of wider criteria. The aim was to ensure suitable judgements and balance of factors relevant to needs of nutritional surveillance programmes and to provide for ongoing quality and detail of food composition data. The nutrient comparisons were conducted using systematic approaches. The ingredient and nutrient profile of previously coded recipes were analysed in a way to match various mixed dishes to generic mixed dish codes in the rationalised NDB. The steps taken for this rationalisation exercise may guide future researchers who wish to create or update food databases used in dietary assessment and highlights awareness and provides resolution for some of the associated complexities in a systematic way.

Koch et al. [8] describes the development of the food database as one of the most challenging parts in the adaptation of the 24 h recall system. The rationalisation of the UK NDB was a complex and time-intensive task including many components, i.e., data review, cleaning, food matching, code revision and evaluation. The process was undertaken over a period of 2 years in parallel with other work, by a team of research assistants with a nutrition background, alongside senior food composition and survey experts, a statistician and a database manager. 

Given the purpose of the NDNS RP to monitor UK population adherence to nutritional recommendations and to evaluate change over time in population dietary intake, following implementation of Intake24, it was important to comprehensively and specifically evaluate the impact of changing the survey dietary assessment method including different components of change [27]. Understanding the specific impact of the rationalisation exercise on NDNS dietary intake data was therefore a key component to consider. From a review of literature prior to undertaking this study, we identified only one other study that had isolated and evaluated the impact of changes in underpinning food lists on resulting data. Evans’ study [17] conducted a very similar evaluation by applying a concise food list to previously collected Irish National Adult Nutrition Survey (NANS) data (2008–2010) and comparing nutrient estimates derived using the original more extensive food list. As in Evans’ study, our study showed a low level of impact on nutrient estimations using a rationalised NDB compared to an extensive NDB at the population level, providing reassurance for continuation of national trend estimates across the NDNS dietary method change. Between the two studies, percentage difference for energy was smaller in Evans’ study (1% in our study compared to 0.0% in Evans’ study) and macronutrients (between 0.0% and 3.1% our study, between (−) 1.4% and 0.3% in Evans’ study) compared to micronutrients (between 0.3% and 9.6% in this study, between (−) 5% and 5.8% in Evans’ study). Over and above the approach in Evans’ study, our evaluation also looked into the impact on food intake for three key foods (meat, fish, and fruit and vegetable) where higher percentage differences (between −5.0% and 5.7%) were found compared to most nutrients. Although the overall significance was low, both studies observed higher % nutrient difference in vitamin D and E (between 9.6% and 6.2% in this study, between (−) 3.4% and (−) 5.0% in Evans’ study) compared to other nutrients. In the rationalisation for NDNS RP, this was possibly due to reformulation or matching non-fortified fat spreads to fortified fat spreads. This finding has indicated that the inclusion of more varieties of common fortified foods such as fat spreads in the food list may be appropriate to reflect the diversity of fortification levels and to improve data capture of fortified foods and product reformulations. The UK NDB is maintained and updated ongoing in parallel with the NDNS RP to reflect the continually changing food environment and to further improve composition estimates. Since the initial implementation of Intake24 in NDNS RP (Software Version 3: Intake24 [UK] Cambridge 2019), the Intake24 food databases have continued to be further updated, including completion of the NDB rationalisation exercise -as described in this paper- along with updates to the wider food databases (including food details, portion estimation and related data) [at the time of publication, the NDNS RP is using https://intake24.org (Software Version 3: Intake24 [UK] Cambridge 2022, accessed on 27 June 2022)]. Even where ongoing updates may be more routine and not represent such a major overhaul to the database as the rationalisation, they still may warrant ongoing evaluation of their impact on population nutrient intakes.

In our evaluation, single foods, recipes, and sandwiches were matched and their impact on nutrient and food intakes were evaluated separately; to our knowledge, this approach is distinct to other studies. As might be expected due to the difference in the former coding approach (reporting and coding of recipes and constituent ingredients rather than single dishes), this exercise demonstrated that nutrient differences were greater in mixed dishes (e.g., lasagne), compared with sandwiches and single foods (e.g., milk). However, the results of our work have demonstrated that even with a higher difference in nutrient intake for mixed dishes between the extensive and rationalised NDB, the effect on overall dietary intake was not significant. Overall, the results of our study have demonstrated that using generic standard food codes for mixed dishes, instead of requiring respondents to enter recipes themselves, had no significant impact on summary estimates for nutrients and foods at the population level. 

Other automated recall tools such as Myfood24 [28], Myfood24 Germany [8] and ASA24 [4] incorporate more extensive food databases, including brand databases. Management of large datasets can present issues for participant usability such as when searches return long lists of results. The evaluation [8] of Myfood24 Germany (*n* = 11,501) showed that a high number of search results were criticised by the respondents although overall the usability of the tool was good and acceptable. Respondent burden of large food lists in 24 h recalls can also be reduced by optimising search mechanisms [8]. The Intake24 tool will need to continue to be maintained and updated in parallel with the NDNS RP to remain up to date, representative of the UK food system, and attractive to research participant users to facilitate collection of high quality data. Further optimisation of the structure and management of food lists and food composition data, incorporation of food brand databases, and use of advanced technologies (including barcode reading and to further optimise search functionality) have scope to efficiently enable increased choice of food items without compromising respondent burden and automation of food data updates.

Where possible, all available NDNS RP data from 2008–2018 was utilised to inform the rationalisation and subsequent evaluation exercise. There were some limitations in scope due to a number of factors. For example, the evaluation exercise used comparison data from only one year of NDNS RP (year 2017, the most recent data available) for two reasons. Firstly, recipes were more easily identifiable in NDNS year 2017 dataset due to coding approaches. Secondly, the relinking of a large number of foods was largely manual intensive work and required considerable resource and it was not feasible to include other survey years due to resource and time constraints. 

In the NDB rationalisation exercise, not all of the mixed dishes in the rationalised NDB were checked and revised in terms of their composition due to capacity and time pressures. Although in the evaluation exercise, no major differences were observed following the NDNS year 2017 recoding, particularly for sandwiches and single foods, a further review of the composition of codes used for mixed dishes in the rationalised NDB may be beneficial to improve their constituent components. It will also be important to provide sufficient variety of mixed dishes in Intake24 and to maintain and update the NDB accordingly. These observations will be considered and be taken forward to ensure both participant ability to find and report their food with confidence and to provide the detail required for nutritional surveillance. 

Quality checks undertaken as part of the exercise and evaluation suggested that further updates in the rationalised NDB may be beneficial. These were either actioned or noted for future work according to their priority. The main issues observed were due to matching errors that occurred with recipes; this was anticipated due to the estimated basis of matching. Overall, about 621 (0.5%) food items out of 132,051 foods (including repeated food items) consumed in NDNS year 2017 were affected by these issues and the impact was assumed to be minimal (examples on matching errors are provided in Appendix A).

## 5. Conclusions

This paper presents the process for the rationalisation of the UK NDB which was necessary for the implementation of https://intake24.org/ (accessed on 27 June 2022) in the UK NDNS RP for dietary data collection (2019). The rationalisation of the NDB greatly reduced the number of foods in the NDB thereby enabling food databases to be maintained more efficiently ongoing for Intake24 [UK] and NDNS RP. The rationalisation also offers a more concise basis from which to explore the use of other reference food data (e.g., brand databases) to inform on popularity or detail of food products and/or investigating other food resources. Despite the limitations mentioned in the discussion, the results of the evaluation strengthen the evidence of a good agreement between concise food databases and larger food databases incorporated into web-based 24 h recalls for estimating nutrient intakes at the population level. This indicates minimal impact on resulting NDNS RP dietary data and suggests that continuation of the national time series dataset and trend estimates across the dietary method change would be reasonable. The evaluation has also highlighted areas for future improvement for Intake24 and the relevance of ongoing monitoring of the impact of continuous updates in the underlying NDB. The detail of our methodology will be useful to those involved in the management and ongoing development of food databases and/or for their incorporation into web-based/digital dietary data collection systems and to researchers, policy makers and health professionals using NDNS trend data.

## Figures and Tables

**Figure 1 nutrients-14-04551-f001:**
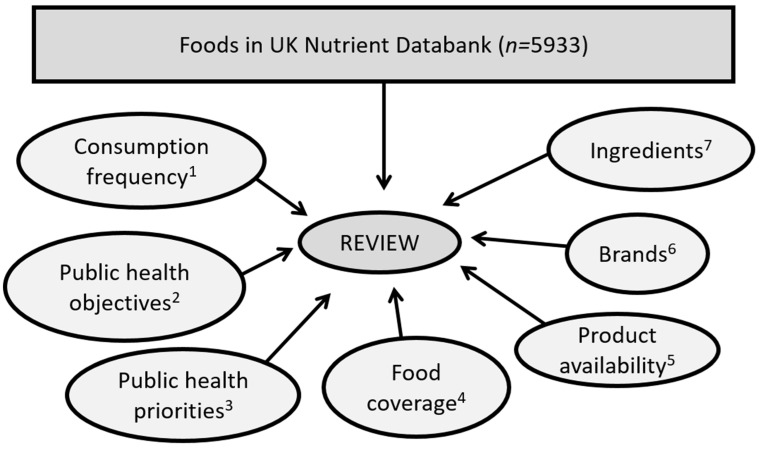
Key factors considered for reviewing foods relevant for inclusion in the rationalised NDB. ^1^ Dietary intake data from the first 10 years of NDNS RP (2008–2018) were examined to identify reported consumption rates. Foods reported fewer than 10 times (average consumption frequency per year) were considered infrequent and mostly excluded from the extensive NDB. ^2^ Detail/specification required for government monitoring objectives for public health and nutritional surveillance, and exposure assessment. ^3^ Relevance to specific UK public health priorities, monitoring and evaluation requirements (e.g., UK sugar reduction policy [14] and exposure assessment). ^4^ Coverage and representation of foods and drinks currently available and consumed in UK, based on the foods available in the extensive NDB and using NDNS Y1-10 consumption rates: retaining foods that could not be adequately represented (either according to their food name or nutrient profile) by other foods existing in the NDB or representing similar foods with a similar nutrient composition profile by another food code. ^5^ Availability of products in the UK market: removing foods that no longer exist in UK food market. ^6^ Popularity of food brands: retaining very popular brand-specific foods (e.g., Toblerone chocolate bar) to effectively monitor the composition of these foods and capture the change in their reformulations. ^7^ Foods that are common ingredients for homemade recipes in the UK diet (e.g., stock cubes).

**Table 1 nutrients-14-04551-t001:** Differences in the mean daily intakes of energy, macronutrients and selected foods from food sources (excluding nutritional supplements) in NDNS year 2017 respondents (*n* = 1211) aged 1.5–93 years using the extensive and rationalised NDB only for replacements of single foods, recipes (e.g., complex dishes) and sandwiches.

	Baseline ^1^ (*n* = 132,052, 100%)	Single Food Replacement only (*n* = 26,958, 20%)		Recipe ^2^ Replacement only (*n* = 32,693, 25%)		Sandwich Replacement ^3^ (*n* = 5694, 4%)	
Macronutrient	Mean	Mean	Diff.	Mean	Diff.	Mean	Diff.
Energy (kcal)	1610	1606	−4	1637	27	1605	−5
Protein (g)	64.4	65.0	0.6	65.8	1.4	64.7	0.3
Carbohydrate (g)	202	200	−2	202	0	202	0
Total fat (g)	61.6	61.5	−0.1	63.9	2.3	60.8	−0.8
Fibre (g)	16.8	16.7	−0.1	16.7	−0.1	16.9	0.1
Food							
Meat (g)	83.7	84.4	0.7	87.6	3.9	84.3	0.6
Fruit & vegetable (g)	252	252	0	243	−9	250	−2
Fish (g)	16.6	16.7	0.1	17.5	0.9	16.5	−0.1

^1^ Extensive NDB. ^2^ Complex dishes such as soups, pizzas. ^3^ Sandwiches such as cheese sandwich. Diff: Difference of mean compared to baseline mean. *n*: Number of rows (food items) in the food level dataset. %: Proportion of food items in baseline dataset set.

**Table 2 nutrients-14-04551-t002:** Differences in the mean daily intake of energy, nutrients, and selected foods from food sources (excluding nutritional supplements) in NDNS year 2017 respondents (*n* = 1211) aged 1.5–93 years using an extensive NDB and a rationalised NDB food composition database.

	Extensive NDB (*n* = 5933)		Rationalised NDB (*n* = 2481)				
Nutrient	Mean	SD	Mean	SD	% Diff. ^1^	*p*-Value	Cohen’s *d*
Energy (kcal)	1610	497	1627	505	1.1	<0.01	0.03
Energy (kJ)	6776	2088	6845	2119	1.0	<0.01	0.03
Protein (g)	64.4	22.4	66.4	23.3	3.1	<0.01	0.09
Protein (%TE)	16.2	3.90	16.5	3.90	1.9	<0.01	0.08
Carbohydrate (g)	202	64.8	202	64.8	0.0	0.98	0.00
Carbohydrate (%TE)	47.4	7.20	46.9	6.90	−1.1	<0.01	−0.07
Free sugars (g)	46.9	29.4	45.5	28.6	−2.8	<0.01	0.05
Free sugars (%TE)	10.7	5.40	10.3	5.20	−3.7	<0.01	0.07
Total fat (g)	61.6	23.5	62.6	23.5	1.6	<0.01	0.04
Total fat (%TE)	34.2	6.00	34.4	5.60	0.6	<0.01	0.04
Saturated fat (g)	23.2	10.3	23.2	9.90	−0.4	0.62	0.00
Saturated fat (%TE)	12.9	3.60	12.8	3.30	−0.8	<0.01	0.04
Fibre (g)	16.8	7.00	17.0	6.90	1.2	<0.01	0.03
Vitamin B_1_ (mg)	1.39	0.51	1.44	0.53	3.6	<0.01	0.10
Vitamin B_2_ (mg)	1.48	0.69	1.49	0.73	0.7	0.54	0.01
Calcium (mg)	779	305	786	307	0.9	<0.01	0.02
Magnesium (mg)	227	85.0	230	83.4	1.3	<0.01	0.03
Phosphorous (mg)	1095	363	1116	367	1.9	<0.01	0.06
Zinc (mg)	7.23	2.65	7.33	2.77	1.4	<0.01	0.04
Iron (mg)	9.06	3.58	9.19	3.65	1.4	<0.01	0.04
Potassium (mg)	2461	837	2483	836	0.9	<0.01	0.03
Vitamin E (mg)	8.65	3.60	9.19	3.78	6.2	<0.01	0.15
Niacin (mg)	29.9	12.0	30.9	12.5	3.3	<0.01	0.09
Sodium (mg)	1790	695	1796	691	0.3	0.57	0.01
Logged nutrient ^2^	GM	25%–75%	GM	25%−75%	% Diff ^1^	*p*-value	Cohen’s *d*
Vitamin A (µg)	589	379–906	599	391–890	1.7	0.06	0.02
Vitamin D (µg)	2.05	1.31–3.40	2.24	1.45–3.73	9.6	<0.01	0.12
Vitamin B6 (mg)	1.40	1.09–1.81	1.43	1.12–1.83	2.4	<0.01	0.05
Vitamin B12 (µg)	3.95	2.85–5.50	4.06	2.95–5.67	2.7	<0.01	0.05
Folate (µg)	187	140–250	190	143–252	1.4	<0.01	0.03
Vitamin C (mg)	65.4	44.2–102	66.2	46.1–104	1.2	0.05	0.02
**Food**	Mean	SD	Mean	SD	% Diff. ^1^	*p*-value	Cohen’s *d*
Meat (g)	83.7	56.1	88.5	59.8	5.7	<0.01	0.08
Fruit & vegetable (g)	252	175	241	165	−4.4	<0.01	0.06
Fruit & vegetable (portions) ^3^	4.00	2.40	3.80	2.30	−5.0	<0.01	0.09
Fish (g)	16.6	25.0	17.5	27.3	5.4	<0.01	0.03

^1^ Difference: Calculated as the % of difference of the mean intake (rationalised-extensive NDB). ^2^ The variables are analysed on the log scale so the mean is a geometric mean (GM), 25–75% values are provided instead of SD. ^3^ Calculated for respondents aged 11 years and over. %TE: Total energy.

**Table 3 nutrients-14-04551-t003:** Association between estimates of energy, nutrients and selected food intakes from food sources (excluding nutritional supplements) in NDNS year 2017 respondents (*n* = 1211) aged 1.5–93 years using the extensive (*n* = 5933) and rationalised (*n* = 2481) NDB.

Nutrient	Proportion in the Same Tertile (%)	Cohen’s κ
Energy (kcal)	90.1	0.83
Energy (kJ)	90.2	0.85
Protein (g)	88.8	0.83
Protein (%TE)	78.7	0.68
Carbohydrate (g)	91.6	0.87
Carbohydrate (%TE)	78.3	0.67
Free sugars (g)	89.1	0.84
Free sugars (%TE)	85.8	0.79
Total fat (g)	82.7	0.74
Total fat (%TE)	73.4	0.60
Saturated fat (g)	83.9	0.76
Saturated fat (%TE)	78.4	0.68
Fibre (g)	88.3	0.82
Vitamin B_1_ (mg)	83.0	0.74
Vitamin B_2_ (mg)	87.7	0.82
Calcium (mg)	85.6	0.78
Magnesium (mg)	89.7	0.85
Phosphorous (mg)	89.3	0.84
Zinc (mg)	87.2	0.81
Iron (mg)	85.8	0.79
Potassium (mg)	89.5	0.84
Vitamin E (mg)	73.6	0.60
Niacin (mg)	84.6	0.77
Sodium (mg)	79.9	0.70
Vitamin A (µg)	78.8	0.68
Vitamin D (µg)	73.6	0.60
Vitamin B6 (mg)	80.3	0.71
Vitamin B12 (µg)	83.7	0.76
Folate (µg)	83.4	0.75
Vitamin C (mg)	85.0	0.77
Food		
Meat (g)	86.9	0.80
Fruit & vegetable (g)	90.4	0.86
Fruit & vegetable (portions) ^1^	87.3	0.81
Fish (g)	96.8	0.93

^1^ Calculated for respondents aged 11 years and over. %TE: Total energy.

**Table 4 nutrients-14-04551-t004:** Percentage of respondents meeting dietary reference values for selected nutrients [19,20,21], and fruit and vegetables [22] intake from food sources (excluding nutritional supplements) in NDNS year 2017 respondents (*n* = 1211) aged 1.5–93 years.

	% Meeting Recommendations Using Extensive Food Database	% Meeting Recommendations Using Rationalised Food Database	% Classified into the Same Category	Cohen’s κ
Carbohydrate (≥47 %TE)	57.0	54.4	88.5	0.77
Free sugars (≤5 %TE)	11.4	13.6	95.5	0.79
Total fat (≥20 ≤35 %TE)	53.3	54.4	82.6	0.65
Saturated fat (≤10 %TE)	21.5	21.6	89.6	0.69
Fibre (≥15–30 %TE ^1^)	9.2	9.7	97.3	0.84
Fruit & vegetable ^2^ (≥5 portions)	29.0	25.4	95.2	0.88

^1^ Recommendations vary based on age. ^2^ Calculated for respondents aged 11 years and over. %TE: Total energy.

## Data Availability

The NDNS RP dataset and respective UK Nutrient Databanks are available from the UK Data Service. The rationalised NDB is deposited under the NDNS: Diet and Physical Activity—a Follow-up Study during COVID-19 (https://doi.org/10.5255/UKDA-SN-8956-2, accessed on 27 June 2022). The NDNS Year 10 original data and NDB is deposited under the NDNS RP Years 1–11 (https://doi.org/10.5255/UKDA-SN-6533-19, accessed on 27 June 2022). The UK NDB-2021, NDNS Year 4 is deposited under the NDNS RP Years 1–11 (https://doi.org/10.5255/UKDA-SN-6533-19, accessed on 27 June 2022). Enquiries about the recoded Year 10 (evaluation exercise) data should be made to the corresponding author.

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
