# Peer review of "Rationalisation of the UK Nutrient Databank for Incorporation in a Web-Based Dietary Recall for Implementation in the UK National Diet and Nutrition Survey Rolling Programme"

_nutrients, 2022, doi:10.3390/nu14214551_

Round 1
Reviewer 1 Report
Dear Authors,
I find your paper very interesting. It would be interesting in order to set it up:
-To make a literature review section.
-The statistical methodology used has been very simple.
-Expand the conclusions.
- To include in the conclusions, limitations, practical implications and future lines of research.
-The references are scarce.
Best regards,
Author Response
Reviewer comment
Dear Authors,
I find your paper very interesting. It would be interesting in order to set it up:
-To make a literature review section.
-The statistical methodology used has been very simple.
-Expand the conclusions.
- To include in the conclusions, limitations, practical implications and future lines of research.
-The references are scarce.
Authors responses
We are glad to hear that you found our paper interesting. We appreciate your time and we have tried to address your comments as set out below.
-To make a literature review section:
Response: The main aim of the paper is to publish our method, rather than do a literature review. Therefore we believe that a detailed literature review section is not necessary for this paper. We did however scan the literature to identify other similar work to inform development of our own approach and method and we discussed these papers in the discussion section.
-The statistical methodology used has been very simple
Response: The statistical analyses were carefully selected and conducted by our senior statistician and were informed by methods used in another study with a similar objective and context (Evans, 2017*), this is referenced in our paper.
* Evans, K.; Hennessy, Á.; Walton, J.; Timon, C.; Gibney, E.; Flynn, A. Development and evaluation of a concise food list for use in a web-based 24-h dietary recall tool. J Nutr Sci 2017, 6, e46, doi:10.1017/jns.2017.49.
-Expand the conclusions. - To include in the conclusions, limitations, practical implications and future lines of research.
Response: We have amended the conclusion by drawing attention to limitations and future research.
-The references are scarce.
Response: The aim of the paper is to publish the method and to evaluate the impact of this, therefore the focus is given to the method and results. In this manuscript, we carefully referred to the papers (n=28) that are aligned with the scope of the paper and that supported our approach, method and results. In addition, the literature on this topic is quite limited which is mentioned in the paper; this is why the references may seem limited.
Reviewer 2 Report
Thank for the opportunity to this review this important and high quality paper. I believe the methodology and insights offered will improve future nutrition surveys and help advance the use of technology for dietary assessment.
Author Response
Reviewer comment
Thank for the opportunity to this review this important and high quality paper. I believe the methodology and insights offered will improve future nutrition surveys and help advance the use of technology for dietary assessment.
Authors response: Thank you for your time and positive feedback. We are very pleased to get your encouraging comments.
Reviewer 3 Report
The topic is important, and the manuscript is well-written. There are a few comments as follows.
1) Figure 1: The explanation of food coverage is vague. How does it differ from product availability and brands? What data did the authors use to determine food coverage? what criteria did the authors use to determine whether to include foods in the rationalized NDB based on food coverage?
2) File S2, Step 1: "3.6g/100g"---What weight does this "/100g" refer to?
3) File S2, Table S2b: What does the leading "g" in "g/100g" in the second row mean?
Author Response
Reviewer comment
The topic is important, and the manuscript is well-written. There are a few comments as follows.
Authors response: Many thanks for your time and encouraging feedback.
- Figure 1: The explanation of food coverage is vague. How does it differ from product availability and brands? What data did the authors use to determine food coverage? what criteria did the authors use to determine whether to include foods in the rationalized NDB based on food coverage?
Response 1: Thank you. This is now amended and final sentence is as below. In addition, the retain and represent sections, in page 4, should give detailed information on the factors considered.
4Coverage and representation of foods and drinks currently available and consumed in UK, based on the foods available in NDB and NDNS consumption rates: retaining foods that could not be adequately represented (either according to their food name or nutrient profile) by other foods existing in the NDB or representing similar foods with a similar nutrient composition profile by another food code.
2) File S2, Step 1: "3.6g/100g"---What weight does this "/100g" refer to?
Thank you. Amended as; 3.6g/100g of edible portion of foods
3) File S2, Table S2b: What does the leading "g" in "g/100g" in the second row mean?
Thank you. Table amended as adding a foot note: *in 100g of edible portion of foods.